# Recent Advances and Challenges in the Treatment of Rhabdomyosarcoma

**DOI:** 10.3390/cancers12071758

**Published:** 2020-07-02

**Authors:** Shinji Miwa, Norio Yamamoto, Katsuhiro Hayashi, Akihiko Takeuchi, Kentaro Igarashi, Hiroyuki Tsuchiya

**Affiliations:** Department of Orthopedic Surgery, Graduate School of Medical Science, Kanazawa University, Kanazawa 920-8640, Japan; norinori@med.kanazawa-u.ac.jp (N.Y.); hayashikatsu830@aol.com (K.H.); a_take@med.kanazawa-u.ac.jp (A.T.); kenken99004@yahoo.co.jp (K.I.); tsuchi@med.kanazawa-u.ac.jp (H.T.)

**Keywords:** rhabdomyosarcoma, chemotherapy, molecular targeted drug, immunotherapy, therapeutic target

## Abstract

Rhabdomyosarcoma, the most common soft tissue sarcoma noted in childhood, requires multimodality treatment, including chemotherapy, surgical resection, and/or radiation therapy. The majority of the patients with localized rhabdomyosarcoma can be cured; however, the long-term outcomes in patients with metastatic rhabdomyosarcoma remain poor. The standard chemotherapy regimen for patients with rhabdomyosarcoma is the combination of vincristine, actinomycin, and cyclophosphamide/ifosfamide. In recent clinical trials, modifications of the standard chemotherapy protocol have shown improvements in the outcomes in patients with rhabdomyosarcoma. In various type of malignancies, new treatments, such as molecular targeted drugs and immunotherapies, have shown superior clinical outcomes compared to those of standard treatments. Therefore, it is necessary to assess the benefits of these treatments in patients with rhabdomyosarcoma. Moreover, recent basic and clinical studies on rhabdomyosarcoma have reported promising therapeutic targets and novel therapeutic approaches. This article reviews the recent challenges and advances in the management of rhabdomyosarcoma.

## 1. Introduction

Rhabdomyosarcoma (RMS) arises from immature cells with the ability to differentiate into skeletal muscle cells in the future. RMS can arise from soft tissues, such as the skeletal muscle, connective tissue, bone, bladder, prostate, testis, nose, orbit, and anus [1]. Approximately 70% of the patients with RMS are diagnosed before the age of 10 years; however, RMS can also develop in adolescents and adults. It is a relatively rare cancer that accounts for only 5–8% of all childhood malignancies but is the most commonly noted type of soft tissue sarcoma in childhood. RMS is classified as embryonal RMS (ERMS), alveolar RMS (ARMS), pleomorphic RMS (PRMS), and spindle cell/sclerosing RMS (SRMS), based on the histological features [2,3,4]. Chromosomal translocations of *t*(2;13)(q35;q14) or *t*(1;13)(q36;q14) are detected in most patients with ARMS. Approximately 60% of the patients with ARMS express *PAX3-FOXO1* and 20% of those express *PAX7-FOXO1* [5,6]. The PAX-FOXO1 chimeric protein functions as an active transcription factor, leading to oncogenic transformation by inducing the expression of abnormal genes. Previous studies have reported that PAX-FOXO1 fusion proteins have oncogenic potential and function as dominant oncogenes in promoting tumorigenesis in fusion-positive RMS [7,8]. In contrast, specific chimeric genes are not associated with ERMS and the other types of RMS; however, these tumors are often associated with various chromosomal abnormalities, which, in turn, lead to the inactivation of the tumor suppressor gene *p53* pathway [9,10]. Skapek et al. reported that positivity for *PAX3-FOXO1* or *PAX7-FOXO1* significantly correlated with worse event-free survival (EFS) and overall survival (OS) [11]. Based on these reports, *PAX-FOXO1* may be considered a strong potential therapeutic target and biomarker predicting prognosis in RMS patients with *PAX-FOXO1* translocation.

Standard treatment of RMS comprises chemotherapy (vincristine, actinomycin D, and cyclophosphamide/ifosfamide), radiation therapy, and surgical tumor excision. Although most patients with localized RMS can be cured, the outcomes in those with metastatic or recurrent RMS remain poor [12,13]. Approximately 15% of the patients with RMS have metastatic lesions at diagnosis [14], and long-term EFS in patients with metastatic RMS is <20% [15,16,17]. Therefore, there is a need to develop systemic treatments, which have better oncological and functional outcomes with long-term safety, for patients with RMS. Several basic and clinical studies have been carried out on various treatment strategies, including modified or novel chemotherapy protocols, molecular targeted drug therapy, immunotherapy, and new therapeutic approaches for RMS. In this review article, the recent challenges and advancements in the treatment of RMS are discussed.

## 2. Chemotherapy

Patients with RMS require multidisciplinary treatment, including chemotherapy, surgical resection and radiation therapy (RT). Multi-drug chemotherapy regimens significantly improved the outcome in patients with localized RMS, although no marked improvement has been observed in patients with metastatic RMS [18]. Standard chemotherapy regimens for RMS in North America include vincristine, actinomycin D, and cyclophosphamide (VAC), whereas those in Europe include ifosfamide, vincristine, and actinomycin D (IVA) [19,20,21]. In a randomized trial comparing the VAC and IVA regimens, significant differences in the clinical outcomes were not observed [22]. In contrast, high-dose chemotherapy was thought to improve the outcomes compared to standard chemotherapy in patients with metastatic RMS. However, it was reported that a significant improvement was not observed in patients treated with high-dose chemotherapy compared to that in those treated with standard chemotherapy, although an increased incidence of treatment-related adverse events (AEs) was observed in patients with high-dose chemotherapy [23]. Furthermore, high-dose chemotherapy with stem-cell rescue did not show a significant advantage for the treatment of metastatic RMS [24]. In contrast, doxorubicin has been widely used for the treatment of soft tissue sarcomas. However, its role in the treatment of RMS remains controversial. To evaluate the efficacy of the addition of doxorubicin in the standard multidrug regimen, a multicenter randomized controlled phase 3 trial was conducted by the European pediatric Soft tissue sarcoma Study Group (EpSSG; Table 1) [25]. In the study, 484 patients with RMS were randomly assigned to the IVA group and IVA with doxorubicin group. The three-year EFS rates of the IVA group and IVA with doxorubicin group were 63% and 68%, respectively (*p =* 0.33). Severe AEs, including leukopenia, anemia, thrombocytopenia, gastrointestinal disorder, and infection were more commonly noted in the IVA with doxorubicin group. Furthermore, two treatment-related deaths were reported in the IVA with doxorubicin group. Based on the results, the authors concluded that addition of doxorubicin in the standard chemotherapy regimen did not lead to a significant improvement in the clinical outcome of RMS. These studies indicated that the chemotherapy regimen for RMS should be based on the VAC or IVA regimens, and that addition of doxorubicin or high-dose chemotherapy was thought to have little benefit for patients with RMS.

In the pooled analysis of a phase 2 study conducted by the Intergroup Rhabdomyosarcoma Study Group (IRSG) and the Children’s Oncology Group (COG) Soft Tissue Sarcoma Committee, the clinical outcomes of multiagent regimens were compared in patients with high-risk RMS [26]. The overall response rates were 52% for patients who received ifosfamide/doxorubicin (ID); 41%, those who received ifosfamide/etoposide (IE); 55%, those who received vincristine/melphalan (VM); 49% those who received topotecan; 50%, those who received topotecan/cyclophosphamide (TC); and 45%, those who received irinotecan. The disease control rates in the ID, IE, and VM group were higher than those in the topotecan, TC, and irinotecan group. These results indicated that the addition of topotecan or irinotecan to the chemotherapy regimen had no benefit for the management of RMS.

Recently, it was reported that the addition of low-dose maintenance chemotherapy after standard chemotherapy improved the outcome in patients with RMS (Table 1). In a phase 3 clinical trial, 371 patients with high-risk RMS were randomly assigned to standard chemotherapy group and standard chemotherapy with maintenance chemotherapy (low-dose of vinorelbine and cyclophosphamide) group (NCT00339118) [27]. The five-year disease-free survival (DFS) rates were 78% for the patients with maintenance chemotherapy and 70% for the patients without maintenance chemotherapy. The five-year OS rates for the patients with and without maintenance chemotherapy were 87% and 74%, respectively (*p* < 0.001). This study indicated that the addition of maintenance chemotherapy to standard chemotherapy can improve the outcome in patients with high-risk RMS.

A clinical trial was conducted to assess the efficacy of a modified chemotherapy regimen for reducing treatment-associated toxicity in RMS patients. In a phase 3 study conducted by the COG, the efficacy of substituting VAC by vincristine and irinotecan (VI) for half of the duration of chemotherapy was assessed in 448 patients with intermediate-risk RMS (Table 1) [28]. In that study, the cumulative doses of cyclophosphamide in the VAC group and VAC/VI group were 16.8 g/m^2^ and 8.4 g/m^2^, respectively. Although a significant improvement in the oncological outcome in VAC/VI group was not observed, the VAC/VI group showed less severe hematologic toxicity. These results of the study suggest that the VAC/VI regimen is a candidate for alternative standard therapy in patients with intermediate-risk RMS.

Trabectedin, a synthetic alkaloid isolated from the marine ascidian, *Ecteinascidia turbinate*, reduces disease progression and mortality rates for various types of soft tissue sarcoma [29]. In a phase 2 study of trabectedin conducted by the COG, 23 patients with RMS, 16 with Ewing’s sarcoma, and 11 with other soft tissue sarcomas underwent trabectedin treatment [30]. Among the 40 evaluable patients, one patient had a partial response (PR), three patients had stable disease (SD), and 36 patients had progressive disease (PD). This result suggested that trabectedin monotherapy was not sufficient to control recurrent sarcomas in patients with RMS, Ewing sarcoma, and other sarcomas. However, trabectedin is commonly used as a second-line and subsequent chemotherapeutic agent for soft tissue sarcoma, and additional clinical research is required to assess its usefulness in patients with RMS.

Although these recent clinical trials did not report significant improvements in the outcomes of RMS, gradual improvement has been noted on modification of the standard chemotherapy regimens. Additional clinical studies are required to improve the outcomes in patients with RMS, especially in patients with metastatic RMS.

## 3. Molecular Targeted Drugs

While conventional anticancer drugs destroy not only cancer cells, but also normal cells, molecular targeted drugs are thought to specifically attack cells with target molecules involved in growth and proliferation of cancer cells. These molecules, including insulin-like growth factor 1 receptor (IGF-1R), platelet-derived growth factor receptor (PDGFR), vascular endothelial growth factor (VEGF), anaplastic lymphoma kinase (ALK), mesenchymal-epithelial transition factor (MET), and mammalian target of rapamycin (mTOR), are considered the candidates for molecular targets in patients with RMS [18]. To reduce the incidence of treatment-related AEs and to improve outcomes in patients with soft tissue sarcomas, including RMS, clinical trials for various molecular targeted drugs had been conducted (Table 2).

Pazopanib, a multitargeted tyrosine kinase inhibitor, specifically inhibits VEGF receptor (VEGFR), PDGFR, and c-kit [31]. In a phase 2 trial of pazopanib in patients with soft tissue sarcomas conducted by the European Organization for Research and Treatment of Cancer-soft tissue and bone sarcoma group (EORTC), prolonged progression-free survival (PFS) and OS were observed compared to those for the controls [31]. Based on the results, the EORTC conducted a phase 3 trial in 372 patients with STS (NCT00753688) [32]. The study patients with advanced soft tissue sarcomas were randomly assigned into the pazopanib (800 mg, once daily) and placebo groups. In the study, mean PFS was 4.6 months in the pazopanib group and 1.6 months in the placebo group (*p* < 0.0001). However, the OS was 12.5 months in the pazopanib group and 10.7 months in the placebo group, and there was no statistically significant difference between the two groups (*p =* 0.25). Currently, pazopanib has been widely used for the treatment of soft tissue sarcomas, including RMS, although the efficacy in patients with RMS remains unclear. To assess the efficacy of pazopanib for the treatment of RMS, further clinical studies in RMS patients are needed.

VEGF plays an important role in regulating physiologic angiogenesis, and tumor cells induce pathologic angiogenesis via production of VEGF [33]. Therefore, VEGF is considered a therapeutic target in cancer. The efficacy of the anti-VEGF monoclonal antibody, bevacizumab, has been investigated in various malignancies [34,35,36]. In a phase 1 study of bevacizumab in combination with topoisomerase 1 inhibitor, irinotecan, for recurrent/refractory pediatric solid tumors, the PR, SD, and PD rates were 30% (3/10), 30% (3/10), and 40% (4/10), respectively [37]. In the study, one of nine assessable patients (11%) had grade 4 toxicity involving neutropenia and thrombocytopenia. These results suggest that bevacizumab combined with irinotecan seems to be well-tolerated and has an antitumor effect in a proportion of patients with pediatric malignancies.

Sorafenib is a multiple kinase inhibitor of the C-, B-RAF, VEGF-2,3, PDGFR-β, FLT3, and c-KIT signaling pathways [38]. Maruwge et al. reported that sorafenib inhibited tumor growth in an in vitro study and that sorafenib inhibited tumor growth and angiogenesis in a xenograft model of RMS [39]. In a phase 1 trial of sorafenib (200 mg/m^2^) for childhood solid tumors, including RMS and Wilms tumor (NCT01502410), an objective response was not observed [40]. Another phase 2 study on sorafenib (200 mg/m^2^, twice a day) conducted by the COG in 21 patients with refractory solid tumors did not report an objective response in the study patients [40]. In contrast, a phase 2 trial of sorafenib (400 mg, twice a day) involving 101 patients with soft tissue sarcomas showed that 14% of the patients had PR and 33% of those had SD [41]. Although sorafenib showed clinical benefits in adult patients, clinical effects were not observed in pediatric patients. Reconsideration of the dose and frequency of sorafenib may improve the outcomes in pediatric patients.

Crizotinib is a tyrosine kinase inhibitor targeting ALK, MET, ROS proto-oncogene 1 receptor tyrosine kinase (ROS1), and RON [42,43,44]. In a multinational phase 2 clinical study of crizotinib for advanced ARMS (EORTC90101, NCT01524926), the disease control rate was 14%; median PFS, 1.3 months; and median OS, 5.6 months [45]. In the study, the AEs noted were fatigue (39%), nausea (31%), anorexia (31%), vomiting (15%), and constipation (15%). The results of the study indicated that the clinical benefits of crizotinib can be expected in only a small proportion of the patients with ARMS.

Since Akt and mammalian target of rapamycin (mTOR) regulate cell metabolism, growth, proliferation, and survival [46,47,48], these molecules are considered candidates for therapeutic targets for cancers. Temsirolimus, a specific inhibitor of mTOR, can regulate cell growth, proliferation, and survival in cancer cells [49,50,51]. In a basic research study, mTOR inhibitor showed antitumor activity by inhibition of angiogenesis in xenograft models of RMS [52]. In a phase 2 study of temsirolimus, 52 patients with glioma, neuroblastoma, or RMS were treated with 75 mg/m^2^ temsirolimus once weekly [53]. In the study, the disease control rates in patients with high-grade glioma, neuroblastoma, and RMS were 41% (7 of 17 patients), 32% (6 of 19 patients), and 6% (1 of 16 patients), respectively. In another phase 1 study on combination therapy using perifosine and temsirolimus in patients with pediatric solid tumors, 8 of 19 patients had SD and 11 patients had no objective response [54]. In the study, the most commonly noted treatment-related toxicities (grade 3 or 4) were thrombocytopenia (38%), neutropenia (24%), lymphopenia (24%), and hypercholesterolemia (19%). Although these studies suggested that temsirolimus-based treatments are feasible and safe, the response to the treatment is limited in patients with RMS.

IGF-1R, which is strongly expressed in RMS and is associated with tumor initiation and progression [55,56], is considered a potential therapeutic target. In basic studies, cixutumumab, a human monoclonal antibody against IGF-1R, showed antitumor activity in vitro and in vivo [57,58]. In a phase 2 study of cixutumumab conducted by the COG, patients with refractory solid tumors underwent treatment with cixutumumab (9 mg/kg, intravenous, weekly) (NCT00831844) [59]. The study included 102 patients with osteosarcoma, RMS, neuroblastoma, Wilms tumor, adrenocortical carcinoma, hepatoblastoma, and synovial sarcoma. Among these patients, 4 of 20 patients with neuroblastomas and 1 of 20 patients with RMS showed PR, and SD was observed in 14 patients (six patients with neuroblastomas, three patients with RMS, two patients with synovial sarcoma, one patient with Wilms tumor, one patient with osteosarcoma, and one patient with adrenocortical carcinoma). Another study conducted by the COG investigated the efficacy of adding cixutumumab or temozolomide to the standard chemotherapy regimen in 168 patients with metastatic RMS [60]. In this study, an improvement in the outcome in response to the added cixutumumab or temozolomide was not observed. In another phase 2 study on combination treatment with cixutumumab and temsirolimus in 43 patients with recurrent or refractory sarcoma, an objective response was not observed [61]. These results indicated that cixutumumab seems to have only a small effect on the oncological outcomes in patients with advanced RMS. Because RMS is only a small population of these studies, it is difficult to assess the efficacies of the agents in patients with RMS. To assess the safety and efficacy of these treatment in patients with RMS, further clinical trials dedicated to RMS are demanded.

As PAX-FOXO1 is expressed in 80% of the tissues with ARMS but not in the normal tissues, the chimeric transcription factor is considered a promising target in patients with ARMS. In a basic research, lipid-prostamine-siRNA (LPR) nanoparticles targeting *PAX-FOXO1* inhibited the production of the fusion transcript and proliferation of RMS cells [62], and the LPR nanoparticles targeting *PAX-FOXO1* significantly inhibited tumor growth in a xenograft model of ARMS. In contrast, Bharathy et al. reported that entinostat, a class-I specific histone deacetylase inhibitor, inhibited production of the PAX3-FOXO1 fusion protein [63], and that entinostat induced sensitization to chemotherapy by destabilization of *PAX3-FOXO1* mRNA in RMS cells in vitro and in vivo. These basic studies reported the potential of *PAX-FOXO1* as a therapeutic target in RMS. Further preclinical studies on *PAX-FOXO1*-targeting treatment are needed to assess the usefulness of *PAX-FOXO1* as a therapeutic target for RMS.

## 4. Radiation Therapy

RT is one of the standard treatment modalities used in the management of RMS [64,65]. Patients with RMS are stratified into the low, intermediate, and high-risk groups according to the tumor location, size, histological subtype, involvement of lymph node, metastatic lesion, and surgical margin [66]. Although patients with low-risk RMS who undergo excision with wide margins do not require RT, most RMS patients require RT, and the radiation doses are determined according to the risk. Patients with low-risk RMS with gross tumor excision and positive surgical margins require a radiation dose 36 Gy during RT, whereas those with positive nodes require 41.4 Gy and those with gross residual tumor require 50.4 Gy. Wolden et al. investigated local control after standard chemotherapy and RT in patients with RMS [67]. In their study, the clinical outcomes in 423 patients with ARMS (group I-II: *n =* 41; group III: *n =* 102) and group III ERMS (*n =* 280) were analyzed. The five-year EFS and local failure rates in patients with group I/II ARMS were 69% and 10%, whereas the five-year EFS and local failure rates in patients with group III RMS were 70% and 19%, respectively. This study indicated that excellent local control could be achieved with chemotherapy combined with RT in patients with intermediate-risk RMS.

RT induces various complications, such as dermatitis and secondary cancer caused by normal tissue damage; therefore, long-term safety of the treatment is essential in pediatric patients with malignancies. Recently, intensity modulated radiation therapy (IMRT) and proton RT have been commonly performed to concentrate the radiation on the target lesion and to avoid irradiation of the normal cells. Lin et al. investigated the effect of IMRT and three-dimensional conformal radiotherapy (3D-CRT) in 375 patients with intermediate-risk RMS [68]. The study showed that IMRT improved the target coverage compared with 3D-CRT, although no improvement was observed with respect to local control in the study patients. In contrast, a physical characteristic of the proton beam is that it emits most of the energy near the end of its reach. This characteristic makes it possible to concentrate the proton beam on the target lesion. Since proton RT can reduce the toxicity of RT by prevention of damage to normal tissues [69,70], proton RT can be expected to reduce the incidence of the RT-related complications. A phase 2 study comparing proton RT and IMRT in pediatric RMS showed that the integral dose in IMRT was 1.8 times higher than in proton RT, and significant normal tissue sparing was noted with proton RT, such that 87% of essential structures were spared [71]. The results suggest that proton RT can reduce the incidence of RT-related AEs. In a phase 2 study on proton RT in 57 patients with metastatic ERMS treated with chemotherapy, the five-year EFS, OS, and local control rates were 69%, 78%, and 81%, respectively [72]. In the study, 11 of 57 (13%) patients presented with acute grade 3 treatment-related toxicities, including odynophagia, dermatitis, dry eye, mucositis, otitis, and hepatopathy. In contrast, 3 of 43 (7%) patients presented with late treatment-related toxicities, including cataract, chronic otitis, and retinopathy. Proton RT is considered beneficial for pediatric patients with malignancies. Additional investigations of the clinical outcomes and long-term safety of proton RT in patients with RMS are warranted to determine the efficacy of proton RT.

## 5. Immunotherapy

### 5.1. Immune Checkpoint Inhibitors

According to several basic and clinical studies, the immune checkpoint axis is considered a strong therapeutic target in various malignancies. Although the clinical benefits of immune checkpoints as therapeutic targets in soft tissue sarcomas are unclear [73,74,75,76,77], studies have reported on the importance of immune checkpoints in sarcomas. Pollack et al. investigated the expression of programmed death-ligand 1 (PD-L1) and programmed cell death protein (PD-1) in sarcomas, including undifferentiated pleomorphic sarcoma (UPS), leiomyosarcoma, well-differentiated/dedifferentiated liposarcoma, myxoid/round cell liposarcoma, and synovial sarcoma [78]. In the study, the expression of PD-L1 and PD-1 was high in UPS and leiomyosarcoma, while the expression was low in myxoid/round cell liposarcoma and synovial sarcoma. In contrast, Kim et al. investigated the association of PD-L1 expression with prognosis in patients with soft tissue sarcomas, including 32 RMS, 19 synovial sarcomas, 18 Ewing sarcomas, seven epithelioid sarcomas, and six mesenchymal chondrosarcomas [79]. Among the study patients, 38% of those with RMS, 53% of those with synovial sarcoma, 33% of those with Ewing sarcoma, 100% of those with epithelioid sarcoma, and 0% of those with mesenchymal chondrosarcoma showed positive PD-L1 expressions. In the study, the five-year OS rates in patients with PD-L1(+) sarcomas and PD-L1(-) sarcomas were 48% and 68%, respectively (*p* = 0.015). Multivariate analysis revealed a significant association between PD-L1 expression and poor OS. Therefore, the immune checkpoints are considered promising therapeutic targets in patients with RMS.

Only a few clinical studies have investigated the efficacy of immune checkpoint inhibitors for patients with RMS. Davis et al. investigated the efficacy and safety of nivolumab in a phase 1/2 trial in patients with solid tumors and lymphoma (NCT02304458) [80]. In the study, 85 patients (22 neuroblastomas, 22 lymphomas, 12 RMSs, 11 Ewing sarcomas, 13 osteosarcomas, and one melanoma, two epithelioid sarcomas, two other sarcomas) were enrolled. Four of 20 (20%) patients with lymphoma showed an objective response, while none of the 74 patients with solid tumors showed an objective response. Among the patients with solid tumors, 33% (11 of 33 patients) of those with sarcomas and 50% (5 of 10 patients) of those with neuroblastomas showed SD. In another phase 1 study of ipilimumab in pediatric patients with solid tumors, the safety and efficacy of ipilimumab were assessed in patients with solid tumors, solid tumors, including 12 patients with melanoma, eight patients with osteosarcoma, three patients with clear cell sarcoma, three patients with carcinoma, two patients with synovial sarcoma, two patients with RMS, one patient with pleomorphic sarcoma, one patient with neuroblastoma, and one patient with undifferentiated sarcoma [81]. In the study, immune-related AEs, including pneumonitis, pancreatitis, endocrinopathies, colitis, and transaminitis were observed at 5 mg/kg and 10 mg/kg dose levels. Among these patients, six patients had SD (melanoma, osteosarcoma, clear cell sarcoma, and synovial sarcoma), and none of the patients presented with PR or complete response (CR). Because these clinical trials enrolled only a small number of RMS patients, it is difficult to assess efficacies of the treatments for RMS.

Currently, only a few clinical studies have reported the efficacy of immune checkpoint inhibitors in patients with RMS. According to the limited number of clinical studies, immune checkpoint inhibitors seem to have little clinical benefit in RMS patients. As the sample population was small, additional studies with a large number of study patients are needed to evaluate the efficacy of immune checkpoint inhibitors.

### 5.2. Cellular Immunotherapy

Although anti-cancer immunity and cancer immunotherapy have recently attracted attention [82], cancer immunotherapy has been administered for a long time. Coley’s toxin, including live or inactivated bacteria, including *Serratia marcescens* and *Streptococcus pyogenes*, was the first immunotherapy reported in 1891 [83]. In that study, 10% of the patients with inoperable sarcoma showed tumor regression after the injection of Coley’s toxin. However, several studies have investigated various immune stimulators, including muramyl tripeptide, *Bacillus Calmette-Guerin* (BCG), and allogenic tumor cells, and have reported no significant improvement in patients with sarcomas [84,85,86,87]. In contrast, recent studies on the tumor microenvironment and antitumor immune system have contributed to the development of immunotherapy [88]. Dendritic cells (DCs), a type of antigen presenting cells (APCs), phagocytize apoptotic cancer cells, process them, and present tumor-associated antigens (TAAs) via major histocompatibility complexes (MHCs). T lymphocytes recognize the TAAs presented on the surface of the DCs. Cytotoxic T lymphocytes detect cancer cells by identifying TAAs and eliminate cancer cells. The antitumor immune responses are induced by delivering co-stimulatory signals among the APCs and effector cells. In normal immune condition, tumor cells with TAA are eliminated by the antitumor immune system. In contrast, early stage of cancer is associated with insufficient functioning of the immune system because of the suppression of immune checkpoints and recruitment of immune-suppressive cells, such as myeloid-derived suppressor cells and regulatory T cells. Based on the immune response and the escape system, these systems are considered promising therapeutic targets for various type of cancers. As DCs play a central role in the antitumor immune system, clinical studies on DC-based immunotherapy for various malignancies have been conducted. In a clinical study on immunotherapy using DCs in combination with standard chemotherapy in 43 patients with metastatic and/or recurrent pediatric sarcomas, 29 patients underwent immunotherapy (Table 3) [89]. In the study, T-cell responses to tumor lysates were observed in 62% (16 of 26 patients) of the patients treated with immunotherapy, and significantly higher survival rates were observed for patients with an immune response than in those without an immune response (73% vs. 37%, *p =* 0.017) [89]. In contrast, Krishnadas et al. conducted a phase 1/2 study on decitabine and DC vaccine targeting the melanoma-associated gene (MAGE) and New York esophageal squamous cell carcinoma-1 (NY-ESO-1) in children with advanced solid tumors (NCT01241162). In the study, patients with relapsed/refractory solid tumors (neuroblastoma, Ewing’s sarcoma, osteosarcoma, and RMS) underwent decitabine treatment with autologous DCs pulsed with NY-ESO-1, MAGE-A1, and MAGE-A3 (Table 3) [90]. The study reported that one of the eight evaluable patients presented with CR, one patient presented with PR, and six presented with PD; the patient with CR presented with complete remission 3.5 years after the treatment. These studies suggest that a proportion of the patients showed a response to DC-based immunotherapy and that T-cell response seems to be important for the clinical outcomes. Adjuvant treatment activating T-cell response may enhance the effect of DC-based immunotherapy. These studies included only a small population of RMS patients, and further studies are demanded to assess the immunotherapies for RMS.

Recently, adoptive T-cell therapy (ACT) has been considered a promising treatment for cancer [88,91,92,93,94]. In a basic research study, T cells were engineered to express chimeric receptors composed of the antigen-binding domain of a human anti-fetal acetylcholine receptor (fAChR) antibody. In the study, the interaction between fAchR-transduced T cells and fAchR-positive RMS cell lines induced T-cell activation by IFN-γ secretion, and delayed lysis of tumor cells was observed [95]. Huang et al. reported that IGF-1R and tyrosine kinase-like orphan receptor 1 (ROR1) were strongly expressed in Ewing sarcoma, osteosarcoma, ARMS, ERMS, and fibrosarcoma cell lines [92]. Furthermore, the adoptive transfer of chimeric antigen receptor (CAR) T cells targeting the IGF-1R and ROR1 significantly reduced tumor growth in xenograft models [92]. In a clinical trial of ACT using T-cells engineered with T-cell receptor (TCR) directed against NY-ESO-1 in patients with melanoma or synovial sarcoma, objective responses were observed in four of six patients with synovial sarcoma and 5 of 11 patients with melanoma [96]. In another phase 1/2 clinical study on T cells expressing human epidermal growth factor receptor 2 (HER2)-specific chimeric antigen receptor in patients with HER2-positive sarcomas, 4 of 17 showed SD (Table 3) [91]. Three of four patients with SD underwent tumor resection, and one of the resected specimens showed ≥90% necrosis. Although only a few clinical studies on ACT have included RMS patients, ACT is considered a promising treatment due to the specificity for TAAs.

## 6. Basic Studies of Novel Therapeutic Approaches

Recent basic studies have reported promising therapeutic approaches, including ferroptotic agents, oncolytic virus, and tumor-targeting bacterial therapy.

Ferroptosis is non-necrotic and non-apoptotic form of programmed cell death which requires abundant intracellular free iron to promote lipid peroxidation and accumulation of reactive oxygen species. Chen et al. reported that RMS had vulnerability to oxidative stress [97]. In the study, a genomic analysis showed that RMS samples had nucleotide mutations associating with oxidative stress, which indicates high levels of oxidative damage [97]. Furthermore, xenograft models of RMS showed sensitivity to several compounds which enhanced oxidative stress such as cerivastatin, auranofin, and carfilzomib. Oxidative damages in RMS suggest that RMS may be sensitive to oxidative stress inducers [98]. Dachert et al. reported that erastin, a glutathione-depleting agent, induced reactive oxygen species production, lipid peroxidation, and ferroptosis [99]. Based on these reports, ferroptotic agents may be promising treatments in RMS, although the further investigations for mechanisms of cell death by ferroptotic agents are demanded.

ERMS mainly develops in response to aberrant activation of receptor tyrosine kinase-mediated RAS signaling cascade. Phelps et al. investigated the function of RAS signaling in regulating the growth and differentiation of ERMS [100]. In the study, RAS knockout in ERMS xenograft models resulted in tumor regression and myogenic differentiation, and recombinant oncolytic myxoma virus targeting RAS significantly reduced tumor growth and improved OS in a xenograft model of ERMS. In another study, antitumor effects of telomerase-specific oncolytic adenovirus on bone and soft tissue sarcomas were observed [101]. These viruses are capable of tumor-specific replication due to induction of the expression of the adenoviral *E1A* and *E1B* genes under the control of the human telomerase reverse transcriptase-encoding gene (hTERT) promoter. Yano et al. reported that oncolytic viruses labelled by green fluorescent protein (GFP) enabled fluorescence-guided surgery based on the specificity for tumor cells [102]. In the study, telomerase-dependent, GFP-containing adenovirus was injected in an orthotopic mouse model of fibrosarcoma. The accumulation of GFP-expressing oncolytic viruses in the tumor tissues was visualized using fluorescence imaging, and fluorescence-guided surgery could be performed by visualization of the tumor tissues during surgical resection.

Zhao et al. developed tumor-targeting *Salmonella typhimurium* (*S. typhimurium*) A1-R [103,104,105,106]. Auxotrophy for Leu-Arg in *S. typhimusrium* A1-R prevents infection in normal tissues; however, *S. typhimusrium* A1-R specifically accumulate in the tumor tissue and virulence. *S. typhimurium* A1-R concentrated and eradicated the primary and metastatic tumors in an in vivo study [105,106]. Igarashi et al. reported that *S. typhimurium* A1-R inhibited significantly regressed tumor growth compared with that in the untreated control in a mouse model of RMS [107]. To the best of our knowledge, although clinical studies have not reported the efficacy of bacterial therapy for RMS, bacterial therapy may potentially be used for the treatment of RMS in the clinical setting.

To minimize the incidence of treatment-related complications due to damage to normal cells, high specificity of tumor-targeting treatment is warranted. The tumor-specific viral or bacterial therapy may possibly reduce the incidence of complications associated with antitumor treatment. Additional studies investigating these treatments are required to evaluate the efficacy and safety of these treatments for patients with RMS.

## 7. Precision Medicine

Expression of some molecules may be associated with the therapeutic responses to the molecular targeted drugs. Therefore, studies addressing the correlation between expression of the target molecules and therapeutic responses to the molecular target drugs are warranted to predict the clinical responses in patients with soft tissue sarcoma.

It is difficult to conduct clinical trials of anticancer drugs for the treatment of RMS owing to the rarity of RMS, its various subtypes, and differences in gene mutations. Recently, genomic profiling has been used in various malignancies, and genomic profiling data have contributed to the determination of the optimal clinical treatment strategies in patients with malignancies [108]. Although genomic profiling is not commonly performed during the treatment of RMS, several studies have reported on the prediction of therapeutic response in patients with RMS.

Casey et al. reported that RMS patients with the *PAX-FOXO1* fusion protein had a lower tumor mutational burden (TMB) than those without the fusion protein, and that high TMB was significantly associated with worse local control, DFS, and OS [109]. TMB is identified as a biomarker predicting response to immune checkpoint inhibitors [108]. However, assessment of TMB does not seem to be beneficial for predicting the response to immune checkpoint inhibitors in patients with STS since a low TMB has been reported in soft tissue sarcoma [110]. In contrast, microsatellite instability (MSI) can be also used as a biomarker to predict the response to pembrolizumab, an immune checkpoint inhibitor [111,112]. However, only 0.2% of the patients with soft tissue sarcomas reportedly showed high-MSI [113]. Further investigations of biomarkers predicting the response to molecular targeted drugs and immune checkpoint inhibitors are needed.

Several studies have reported on the prediction of the response to anticancer agents in patients using animal models of cancer. Although the mouse model of human malignant tumors has been used to investigate the mechanism of tumor progression and to evaluate response to anticancer agents, the mouse model created by subcutaneous injection of cancer cells cannot reflect on tumor progression and response to anticancer agents in the human body. In contrast, it was reported that a patient-derived orthotopic xenograft (PDOX) model created by orthotopic implantation of tumor tissues in immunocompromised mice can mimic the characteristics of cancer in patients [114]. In preclinical studies, PDOX models demonstrated characteristics similar to those of sarcoma in patients, including recurrence, metastasis, invasion, and response to anticancer agents [107]. Due to the similarity to cancer in patients, the PDOX model is considered useful in precision medicine for various malignancies, including RMS [107,114,115]. Further information and clinical studies regarding the correlation of response to anticancer agents between the PDOX models and human patients are needed to utilize the prediction of therapeutic response by PDOX model in patients with RMS.

## 8. Conclusions

Because there are only a small number of clinical trials dedicated to RMS, it is often difficult to discuss the efficacy and safety of new treatments for RMS. However, several clinical trials of molecular targeted drugs and immunotherapy showed efficacies in patients with soft tissue sarcomas, including RMS. On the other hand, the usefulness of modified chemotherapy regimens has been demonstrated in recent clinical trials on RMS. Furthermore, basic research including studies on RMS-specific tumor antigen, oncolytic virus and tumor-targeting bacterial therapy have shown the potential to contribute to the development of therapeutic strategies for patients with RMS. These promising treatments should be assessed in clinical studies in the future.

## Figures and Tables

**Table 1 cancers-12-01758-t001:** Recent clinical trials on chemotherapy in patients with Rhabdomyosarcoma (RMS).

Years	Phase	Chemotherapy Regimen	N	Patients	Clinical Significance	Ref.
2019	III	IVA with or without maintenance chemotherapy (VC)	371	Non-metastatic RMS	Five-year DFS: 77.6% with IVA/VC and 69.8% with IVA (*p =* 0.06)	[27]
					Five-year OS: 86.5% with IVA/VC and 73.7% with IVA (*p =* 0.01)	
2018	III	IVA with or without Dox	484	Non-metastatic RMS	Three-year EFS: 67.5% with IVA/Dox and 63.3% with IVA (*p =* 0.33)	[25]
2018	III	VAC or VAC/VI (substitution for half of VAC course by VI)	448	Intermediate-risk RMS	Four-year EFS: 63% with VAC and 59% with VAC/VI (*p =* 0.51)	[28]
					Four-year OS: 73% with VAC and 72% with VAC/VI (*p =* 0.80) less hematologic toxicities with VAC/VI	
2012	II	Trabectedin	50	Recurrent sarcoma: RMS (23), EWS (16), and other sarcomas (11)	PR 2.5%, SD 7.5%, and PD 90%	[30]

IVA: ifosfamide, vincristine, and dactinomycin; VC: vinorelbine and cyclophosphamide; VAC: vincristine, dactinomycin, and cyclophosphamide; VI: vincristine and irinotecan; RMS, rhabdomyosarcoma; EWS, Ewing sarcoma; DFS: disease-free survival; OS: overall survival; EFS: event-free survival; Dox: doxorubicin; PR, partial response; SD, stable disease; PD, progressive disease.

**Table 2 cancers-12-01758-t002:** Clinical trials of molecular targeting agents for sarcomas.

Years	Phase	Treatment	Target Molecule	N	Patients	Clinical Significance	Ref.
2009	II	Pazopanib (800 mg, daily)	PDGF-α, VEGF-1, 2, 3, c-kit	142	Advanced STS	Disease control rate: 49% for SS, 44% for LMS, and 39% for other sarcomas	[31]
2012	III	Pazopanib (800 mg, daily) or placebo	PDGF-α, VEGF-1, 2, 3, c-kit	372	Metastatic STS	OS: 12.5 months with pazopanib and 10.7 months with placebo PFS: 4.6 months with pazopanib and 1.6 months with placebo	[32]
2015	II	Sorafenib (200 mg/m^2^, oral)	PDGFRs, VEGFRs, MAPK, and c-kit	20	RMS (10) and Wilms tumor (10)	Objective response: 0%	[40]
2013	II	Sorafenib (400 mg, twice, daily for 28 days)	PDGFRs, VEGFRs, MAPK, and c-kit	101	Advanced STS	PR, 14%; SD, 33%; PD, 49%, unevaluable 8% Median PFS: four months Median OS: 12 months	[41]
2018	II	Crizotinib (250 mg, twice, daily)	MET, ALK, ROS1 and RON	13	Advanced/metastatic ARMS	Objective response rate 14%, disease control rate: 14%	[45]
2012	II	Temsirolimus (75 mg/m^2^)	mTOR	52	High-grade glioma (17), neuroblastoma (19), and RMS (16)	Disease control rate: 41% for glioma, 32% for neuroblastoma, and 6% for RMS	[53]
2014	II	Cixutumumab (9 mg/kg, weekly)	IGF-1R	114	Relapsed or refractory solid tumors: osteosarcoma (11), RMS (20), neuroblastoma (30), Wilms tumor (10), Adrenocortical carcinoma (10), Hepatoblastoma (10), and synovial sarcoma (11)	PR: 20% for neuroblastoma and 5% for RMS	[59]

PDGFR, platelet-derived growth factor receptor; VEGFR, vascular endothelial growth factor receptor; MAPK, mitogen-activated protein kinases; ALK, anaplastic lymphoma kinase; ROS1, ROS proto-oncogene 1 receptor tyrosine kinase; RON, Recepteur d’Origine Nantais; IGF-1R, insulin-like growth factor 1 receptor; RMS, rhabdomyosarcoma; STS, soft tissue sarcoma; ARMS, alveolar rhabdomyosarcoma; LMS, leiomyosarcoma; OS, overall survival; PR, partial response; SD, stable disease; PD, progressive disease; PFS, progression-free survival.

**Table 3 cancers-12-01758-t003:** Recent clinical studies on immunotherapy.

Year	Phase	Treatment	N	Patients	Clinical Significance	Ref.
2016	II	Standard therapy with or without immunotherapy using autologous lymphocytes, TL-pulsed DCs, with or without IL-7	43	High-risk pediatric sarcomas: EWS (24), RMS (11), and others (8)	T-cell responses toTL: 62% Five-year OS: 73% in patients with T cell response and 37% in patients without T cell response	[89]
2015	I/II	Decitabine followed by DC pulsed with MAGE-A1, MAGE-A3 and NY-ESO-1 peptides	10	NB (8), EWS (1), and RMS (1)	CR: 10%, SD 20%, PD 70%	[90]
2015	I/II	HER2-specific CAR T cells	19	HER2-positive tumors: Osteosarcoma (16), PNET (1), EWS (1), and DSRCT (1)	SD 24%	[91]

TL, tumor lysate; DC, dendritic cell, IL, interleukin; MAGE, melanoma-associated gene; NY-ESO-1, New York esophageal squamous cell carcinoma-1; HER2, human epidermal growth factor receptor 2; CAR, chimeric antigen receptor; EWS, Ewing’s sarcoma; RMS, rhabdomyosarcoma; NB, neuroblastoma; PNET, primitive neuroectodermal tumor; DSRCT, desmoplastic small round cell tumor; TL, tumor lysate; OS, overall survival; CR, complete response; SD, stable disease; PD, progressive disease.

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
