# Peer review of "Recent Advances and Challenges in the Treatment of Rhabdomyosarcoma"

_cancers, 2020, doi:10.3390/cancers12071758_

Round 1

Reviewer 1 Report

In the paper “Recent advances and challenges in the treatment of rhabdomyosarcoma”, the authors review the treatment strategies currently employed against rhabdomyosarcoma (RMS) and the novel therapeutic approaches and advances in the management of this pediatric cancer. In detail, they focus on chemotherapy, radiotherapy, immunotherapy, molecular targeted drugs and novel therapeutic approaches including precision medicine, describing the identified molecular targets and the clinical trials performed in numerous studies. The presented work is exhaustive and interesting. Some minor points should be addressed to improve it:

  • The wide number of data relative to the percentage of patients involved in clinical trials should be mainly confined to tables. This can help the reader to appreciate more the manuscript.
  • To further improve the work, the use of subparagraphs should be exploited.
  • The abbreviations should be carefully checked, some of them are missing and some are not mentioned at their first appearance in the text.
  • In the paragraph “Basic studies of novel therapeutic approaches”, the authors may describe also the use of ferroptotic agents as new potential treatments whose efficacy and safety should be evaluated in RMS patients.
  • In the paragraph “Precision medicine”, some informations appear redundant, as already listed in “Molecular targeted drugs”.
  • References should be checked; some of them present some errors or appear incomplete (for example, number 10, 106, 107, 112).

Author Response

Dear reviewer, we sincerely appreciate you for your considering our manuscript entitled “Recent advances and challenges in the treatment of rhabdomyosarcoma” for publication in Cancers. We are very grateful for your prompt attention and thorough review. Based on your comments, we have revised the manuscript and have addressed all of the concerns brought up. Please see below for our point-by point responses to each of the comments. If there are any further issues that need to be resolved to improve our manuscript, please let us know.

Newly added or changed parts are indicated in red in the revised manuscript.

Comments:

The wide number of data relative to the percentage of patients involved in clinical trials should be mainly confined to tables. This can help the reader to appreciate more the manuscript.

Response:

Thank you for your recommendation. Information about the number of RMS patients were included in tables.

To further improve the work, the use of subparagraphs should be exploited.

Response:

Thank you for your suggestion. In the manuscript, subparagraphs of “immune checkpoint inhibitors” and “cellular immunotherapy” were created and included in the paragraph of “immunotherapy”.

The abbreviations should be carefully checked, some of them are missing and some are not mentioned at their first appearance in the text.

Response:

Thank you for your pointing this out. The abbreviations were checked and corrected.

In the paragraph “Basic studies of novel therapeutic approaches”, the authors may describe also the use of ferroptotic agents as new potential treatments whose efficacy and safety should be evaluated in RMS patients.

Response:

Thank you for your suggestion.

To discuss the use of ferroptotic agents as new potential treatments, the following sentences and references were added to the paragraph “Basic studies of novel therapeutic approaches”.

Ferroptosis is non-necrotic and non-apoptotic form of programmed cell death which requires abundant intracellular free iron to promote lipid peroxidation and accumulation of reactive oxygen species. Chen et al. reported that RMS had vulnerability to oxidative stress. In the study, a genomic analysis showed that RMS samples had nucleotide mutations associating with oxidative stress, which indicates high levels of oxidative damage. Furthermore, xenograft models of RMS showed sensitivity to several compounds which enhanced oxidative stress such as cerivastatin, auranofin, and carfilzomib. Oxidative damages in RMS suggest that RMS may be sensitive to oxidative stress inducers. Dachert et al. reported that erastin, a glutathione-depleting agent, induced reactive oxygen species production, lipid peroxidation, and ferroptosis. Based on these reports, ferroptotic agents may be promising treatments in RMS, although the further investigations for mechanisms of cell death by ferroptotic agents are demanded.

References

  1. Chen, X.; Stewart, E.;  Shelat, A. A.;  Qu, C.;  Bahrami, A.;  Hatley, M.;  Wu, G.;  Bradley, C.;  McEvoy, J.;  Pappo, A.;  Spunt, S.;  Valentine, M. B.;  Valentine, V.;  Krafcik, F.;  Lang, W. H.;  Wierdl, M.;  Tsurkan, L.;  Tolleman, V.;  Federico, S. M.;  Morton, C.;  Lu, C.;  Ding, L.;  Easton, J.;  Rusch, M.;  Nagahawatte, P.;  Wang, J.;  Parker, M.;  Wei, L.;  Hedlund, E.;  Finkelstein, D.;  Edmonson, M.;  Shurtleff, S.;  Boggs, K.;  Mulder, H.;  Yergeau, D.;  Skapek, S.;  Hawkins, D. S.;  Ramirez, N.;  Potter, P. M.;  Sandoval, J. A.;  Davidoff, A. M.;  Mardis, E. R.;  Wilson, R. K.;  Zhang, J.;  Downing, J. R.;  Dyer, M. A.; St. Jude Children's Research Hospital-Washington University Pediatric Cancer Genome, P., Targeting oxidative stress in embryonal rhabdomyosarcoma. Cancer Cell 2013, 24 (6), 710-24.
  2. Fanzani, A.; Poli, M., Iron, Oxidative Damage and Ferroptosis in Rhabdomyosarcoma. Int J Mol Sci 2017, 18 (8).
  3. Dachert, J.; Ehrenfeld, V.;  Habermann, K.;  Dolgikh, N.; Fulda, S., Targeting ferroptosis in rhabdomyosarcoma cells. Int J Cancer 2020, 146 (2), 510-20.

In the paragraph “Precision medicine”, some informations appear redundant, as already listed in “Molecular targeted drugs”.

Response:

Thank you for your pointing this out.

The following sentences were deleted from the paragraph “Precision medicine”.

Molecular targeted drugs were developed and approved for patients with various malignancies approximately 20 years ago. Target molecules of the drugs for soft tissue sarcoma include VEGF, PDGFR, c-Kit, and mitogen-activated protein kinase (MAPK).

References should be checked; some of them present some errors or appear incomplete (for example, number 10, 106, 107, 112).

Response:

Thank you for your pointing this out.

The references were checked and corrected.

Reviewer 2 Report

This is a well written review serving as a modern update as to the scope of therapeutic interventions trialed for rhabdomyosarcoma. I found it to be an interesting historical perspective. The authors should be commended for the comprehensive nature of the review.

My primary concern with this review has little to do with the actual writing of the authors. Instead, I think that the major limitation of such a review is simply the lack of clinical trials dedicated to rhabdomyosarcoma.

I do have concerns about the way in which some of the studies are references. The authors make reference to many studies that include only a small population of patients with rhabdomyosarcoma. It is highly unlikely we will ever be able to see well done clinical trials of all these therapeutic interventions for rhabdomyosarcoma alone. I would simply remind the authors a small population of 10-15 patients with rhabdomyosarcoma amongst a larger trial is inherently difficult to draw conclusions from and I would caution them to comment more extensively on the limitations therein.

For instance, in discussing IGF1R therapies, the authors make reference to a study of 102 patients with sarcoma, including 20 patients with RMS. This study did report outcomes specifically for RMS. I would simply caution that RMS in and of itself was not the primary focus of the study, thus providing inherent limitations to drawing conclusions. Similarly, only 12 of the patients in the study on nivolumab were with RMS. These numbers are really too small to draw meaningful conclusions from.

As such, I would request a major revision to help address the wording and conclusions drawn from such studies.

Of note, there are a few areas upon which the grammar and syntax can be improved.

Author Response

Dear reviewer, we sincerely appreciate you for your considering our manuscript entitled “Recent advances and challenges in the treatment of rhabdomyosarcoma” for publication in Cancers. We are very grateful for your prompt attention and thorough review. Based on your comments, we have revised the manuscript and have addressed all of the concerns brought up. Please see below for our point-by point responses to each of the comments. If there are any further issues that need to be resolved to improve our manuscript, please let us know.

Newly added or changed parts are indicated in red in the revised manuscript.

Comments:

Instead, I think that the major limitation of such a review is simply the lack of clinical trials dedicated to rhabdomyosarcoma.

I do have concerns about the way in which some of the studies are references. The authors make reference to many studies that include only a small population of patients with rhabdomyosarcoma. It is highly unlikely we will ever be able to see well done clinical trials of all these therapeutic interventions for rhabdomyosarcoma alone. I would simply remind the authors a small population of 10-15 patients with rhabdomyosarcoma amongst a larger trial is inherently difficult to draw conclusions from and I would caution them to comment more extensively on the limitations therein.

For instance, in discussing IGF1R therapies, the authors make reference to a study of 102 patients with sarcoma, including 20 patients with RMS. This study did report outcomes specifically for RMS. I would simply caution that RMS in and of itself was not the primary focus of the study, thus providing inherent limitations to drawing conclusions. Similarly, only 12 of the patients in the study on nivolumab were with RMS. These numbers are really too small to draw meaningful conclusions from.

As such, I would request a major revision to help address the wording and conclusions drawn from such studies.

Response:

We agree with your suggestion.

Most of clinical trials of molecular targeted drugs and immunotherapy enrolled only a small number of RMS patients. In these studies, it is thought to be difficult to assess efficacies and safety of the treatments for RMS. To avoid misunderstanding, the number of RMS patients were added to Tables. Also, the difficulties to conclude the efficacies of the treatments for RMS in these studies were described in each paragraph.

Furthermore, to clarify the limitation, the following sentences were added to the conclusions.

Because there are only a small number of clinical trials dedicated to RMS, it is often difficult to discuss the efficacy and safety of new treatments for RMS. However, several clinical trials of molecular targeted drugs and immunotherapy showed efficacies in patients with soft tissue sarcoma.

Reviewer 3 Report

Overall, this review is well written. It has a comprehensive summary of the
recent advances in the treatment of RMS. It would be better if the authors
can add the years of clinical trials into the tables. Immune checkpoint
inhibitors belong to immunotherapy, so they can be put together. If the
authors can clearly point out the advances of RMS treatments in the Abstract
and Conclusions parts, it would be better for the readers to know them.

Author Response

Dear reviewer, we sincerely appreciate you for your considering our manuscript entitled “Recent advances and challenges in the treatment of rhabdomyosarcoma” for publication in Cancers. We are very grateful for your prompt attention and thorough review. Based on your comments, we have revised the manuscript and have addressed all of the concerns brought up. Please see below for our point-by point responses to each of the comments. If there are any further issues that need to be resolved to improve our manuscript, please let us know.

Newly added or changed parts are indicated in red in the revised manuscript.

Comments:

It would be better if the authors can add the years of clinical trials into the tables.

Response:

Thank you for your recommendation.

The years of clinical trials were added into the tables.

Immune checkpoint inhibitors belong to immunotherapy, so they can be put together.

Response:

Thank you for your suggestion.

In the manuscript, subparagraphs of “immune checkpoint inhibitors” and “cellular immunotherapy” were created and included in the paragraph of “immunotherapy”.

If the authors can clearly point out the advances of RMS treatments in the Abstract and Conclusions parts, it would be better for the readers to know them.

Response:

Thank you for your suggestion.

The conclusions were corrected as follows.

Because there are only a small number of clinical trials dedicated to RMS, it is often difficult to discuss the efficacy and safety of new treatments for RMS. However, several clinical trials of molecular targeted drugs and immunotherapy showed efficacies in patients with soft tissue sarcoma including RMS. On the other hand, the usefulness of modified chemotherapy regimens has been demonstrated in recent clinical trials on RMS. Furthermore, basic research including studies on RMS-specific tumor antigen, oncolytic virus, tumor-targeting bacterial therapy, have shown the potential to contribute to the development of therapeutic strategies for patients with RMS. These promising treatments should be assessed in clinical studies in the future.
